# Acetone as Artifact of Analysis in Terpene Samples by HS-GC/MS

**DOI:** 10.3390/molecules27186037

**Published:** 2022-09-16

**Authors:** Sytze Elzinga, Jorge Dominguez-Alonzo, Raquel Keledjian, Brad Douglass, Jeffrey C. Raber

**Affiliations:** 1The Werc Shop, Company, Monrovia, CA 91016, USA; 2pH Solutions Laboratory, Company, Monrovia, CA 91016, USA

**Keywords:** cannabis, terpenes, terpenoids, entourage effect, acetone, residual solvents, caryophyllene, terpinolene, myrcene, headspace

## Abstract

Cannabis-infused product manufacturers often add terpenes to enhance flavor. Meanwhile, labeling requirements for these same products necessitate testing for residual solvent levels. We have found that heating terpene samples containing an oxygen or air atmosphere results in the detection of significantly higher levels of acetone when compared to the same compound in argon atmosphere using temperature regimes common to headspace autosampler routines. This formation was statistically significant (*p* = 0.05) for most of the predominant terpenes found in cannabis. The largest increase in acetone formation was seen for terpinolene which showed an 885% increase in oxygen atmosphere (4603.6 PPM) when compared to analysis under argon (519.9 PPM). Cannabinoids were shown to reduce this formation and explain why high levels of acetone are not reported in cannabis extracts, even though these can contain up to 40% terpenes.

## 1. Introduction

Terpenes and other volatiles produced by cannabis plants impact the aroma and flavor of cannabis. They are also believed to impact the complicated pharmacology of cannabis and cannabis products. The pharmacological interplay of cannabis constituents other than the primary cannabinoids is commonly referred to as the “Entourage Effect” [1] and is generally considered a positive attribute. In contrast, chemical solvents that are residual in final products leftover from manufacturing processes are considered negative attributes with testing to maximum allowable levels encouraged by the United States Pharmacopeia [2].

Labeling requirements for manufactured cannabis products seek to convey information about active ingredients (terpenes) and contaminants (solvents) in the end product to communicate aspects of purity, activity, hazard, and value to the consumer. These are key aspects of the circular bioeconomy that differentiate natural products, such as essential oils. Yet, extant analytical methods often inflate the presence of solvent signals and deflate the presence of valued ingredients despite a plethora of published methods [3,4].

In this work, we draw attention to the existence of this problem to prevent misanalysis. In addition, we have sought to offer clues to new approaches for formulators of essential oil products to stabilize products intended for human consumption in the presence of oxygen.

We demonstrate that detected acetone signals are a common artifact of terpene degradation in the presence of oxygen. The structures of the investigated terpenes can be found in Figure 1.

## 2. Results

### 2.1. Experiment 1: The Influence of Atmosphere in the Headspace Vial

In the first experiment, samples of pure compounds—as well as two mixtures—were analyzed in an argon, air, or oxygen atmosphere. This analysis was performed on three separate days. The average and the 95% confidence intervals can be found in Figure 2. With the notable exception of α-pinene, a higher amount of acetone is found in samples analyzed in air or oxygen versus samples containing an argon-enriched atmosphere. In all cases—except for β-pinene in argon vs. air— the difference is significant at a 95% level (see error bars). However, we found no significant difference between the samples purged with oxygen versus the samples purged with air alone.

### 2.2. Experiment 2: The Influence of Antioxidants

To investigate the influence of oxygen a second experiment was conducted. In this experiment, the blend with equal ratios of terpenes was used and a small amount of either BHT, tocopherol, cannabidiol (CBD), cannabidiolic acid (CBDA), tetrahydrocannabinol (THC), or tetrahydrocannabinolic acid (THCA) was added. This experiment was conducted in duplicate, and the average and the 95% confidence interval are plotted for acetone (Figure 3) and methanol (Figure 4). This data shows that the addition of the mentioned compounds results in a significant reduction in the amount of acetone that is detected. The data was significant at a 95% level for all compounds. A similar decrease can be found in the amount of methanol that is detected and was significant at the 95% level for all compounds except BHT.

## 3. Discussion

This study presents evidence for the formation of acetone from all the tested terpenes and blends, except for α-pinene, when heated in a headspace sampler at 140 °C for 40 min. The amount of acetone that forms is dependent on the terpene that is heated in the presence of oxygen. The effect has low reproducibility, but is enough to result in statistically significant differences compared to samples that were heated under argon. In addition, heating certain terpenes in the presence of oxygen can result in the formation of other residual solvents (Appendix A). For example, linalool resulted in a statistically significant increase in methanol from 44 PPM under argon to 651 PPM in air.

Small amounts of acetone could be detected in samples under argon. It is possible that this is caused by the presence of residual oxygen in the sample vials since the purging and sealing of headspace vials was performed on the lab bench and not in a sealed and purged environment such as a glovebox. Oxygen levels inside the vials were not monitored in our experimental set-up.

The formation of acetone from terpenes has been previously reported in various atmospheric research papers. A study by Reissell et al. observed the formation of acetone through OH radical and O_3_ initiated reactions in gas phase from 12 different terpenes, including limonene, myrcene, a-pinene, β-pinene, and terpinolene [5]. This reaction was reported to occur at room temperature and at atmospheric pressures with the researchers proposing various reaction schemes from terpenes that result in acetone formation. We speculate that similar processes are responsible for the formation of acetone in this experiment.

Finally, we report that some additives—such as known scavengers of reactive oxygen species and multiple cannabinoids—can decrease or eliminate terpene degradation under the same conditions. So-called “antioxidant” activity of multiple cannabinoids has recently been reported and offers support for this mechanism of action [6]. We suggest considering this protective activity, especially from cannabinoids, a corollary of the Entourage Effect.

## 4. Materials and Methods

### 4.1. Materials and Reagents

Residual solvent, terpene, and cannabinoid reference standards were obtained from LGC (Manchester, NH, USA). Food-grade terpenes (β-caryophyllene, humulene, d-limonene, linalool, myrcene, α-pinene, β-pinene, terpinolene) were acquired from Vigon International (East Stroudsburg, PA, USA).

Argon and compressed air with a purity of >99.998% was acquired from Praxair (Danbury, CT, USA).

A BOOST oxygen canister (Milford, CT, USA) was the source for the oxygen used in experiments.

Antioxidants (butylated hydroxytoluene and tocopherol) were acquired from Sigma Aldrich (St. Louis, MO, USA). A commercially available terpene blend which mimics the natural ratios of terpenes in the cannabis cultivar “Jack Herer” was acquired from The Werc Shop (Monrovia, CA, 91016). CBD, CBDA, THC, and THCA were acquired as isolates from a licensed dispensary in California.

### 4.2. Instrumentation and Analytical Conditions

Residual solvents were analyzed by an external ISO 17025 accredited laboratory utilizing headspace gas chromatography using a mass spectrometer for detection (GC-2010 chromatograph, QP-2010 SE Mass Spectrometer, HS-20 static headspace sampler, Shimadzu, USA). Method parameters can be found in Table 1. The method employed a solventless fully evaporative sample introduction using the headspace sampler. This method was validated at the laboratory and accepted for use by the California Bureau of Cannabis Control. The analytes in this method and their limits of detection, limits of quantification, and action limits set by the California Bureau of Cannabis Control (BCC) can be found in Table 2.

### 4.3. Data Handeling and Statstics

Data was processed digitally (Excel, Microsoft, Redmond, WA USA). All statistical interpretations were performed at the 95% confidence level (*p* = 0.05).


**Experiment 1: The influence of atmosphere in the headspace vial**


For each sample, an empty headspace vial and cap were massed. A 35.0 µL sample of pure compounds or mixture was transferred to the vial. A tube connected to an argon tank with regulator was used to purge the vial for 30 s at a flowrate of 23.6 L per minute. Depending on the sample, the vial was either immediately sealed or backfilled with air or oxygen and immediately sealed. The mass of the sealed vial was recorded and the amount of sample inside the vial was calculated. This procedure corrects for any sample that might have evaporated during the purge procedure. By purging all samples with argon first, we eliminate the possibility that the argon purge step is responsible for the removal of residual solvents that might be present in the samples and thus leading to an erroneous conclusion about the impact of atmosphere. All samples were analyzed by HS-GC/MS for residual solvents. This procedure was repeated on three different days.


**Experiment 2: The influence of antioxidants**


A 35.0 µL sample of a mixture containing β-caryophyllene, humulene, D-limonene, linalool, myrcene, α-pinene, β-pinene, and terpinolene at 12.5% each was transferred into a headspace analysis vial. Depending on the researched condition, either one drop of butylated hydroxytoluene (BHT), one drop tocopherol or 20 mg of a cannabinoid was added. A duplicate was prepared for each sample. No purging was performed on these samples. The vial was immediately sealed and analyzed for residual solvents using HS-GC/MS.

## 5. Patents

A patent application has been filed for chemical additives that stabilize individual terpenes and mixtures thereof.

## Figures and Tables

**Figure 1 molecules-27-06037-f001:**
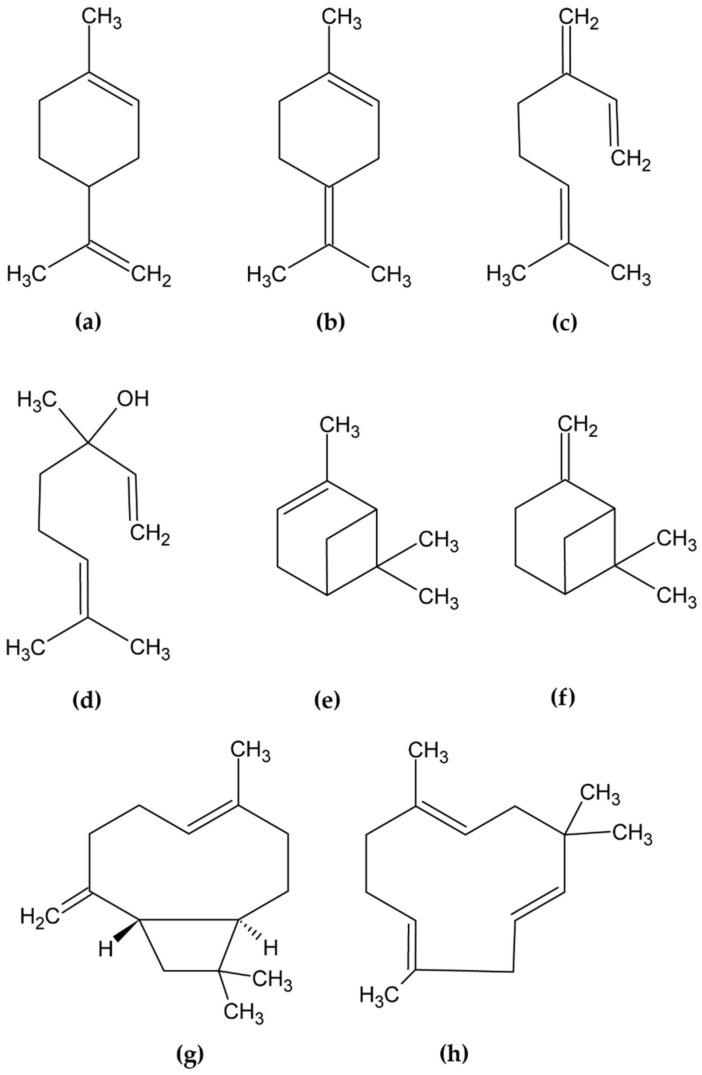
Structures of the investigated compounds: (**a**) limonene, (**b**) terpinolene, (**c**) myrcene, (**d**) linalool, (**e**) α-pinene, (**f**) β-pinene, (**g**) β-caryophyllene, (**h**) humulene.

**Figure 2 molecules-27-06037-f002:**
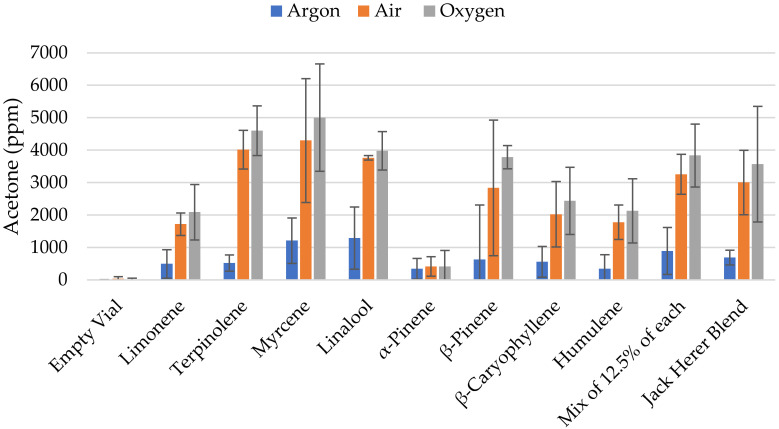
Acetone detection in samples (error bars show the 95% confidence interval).

**Figure 3 molecules-27-06037-f003:**
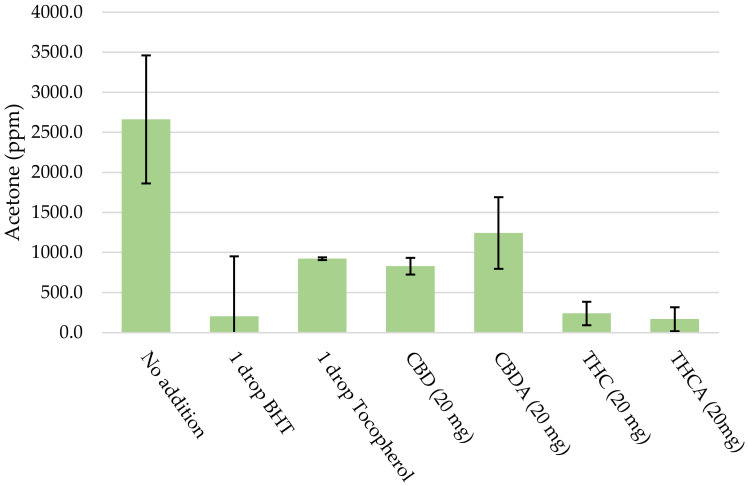
Detection of acetone in samples (PPM) when antioxidants or cannabinoids are present (error bars show the 95% confidence interval).

**Figure 4 molecules-27-06037-f004:**
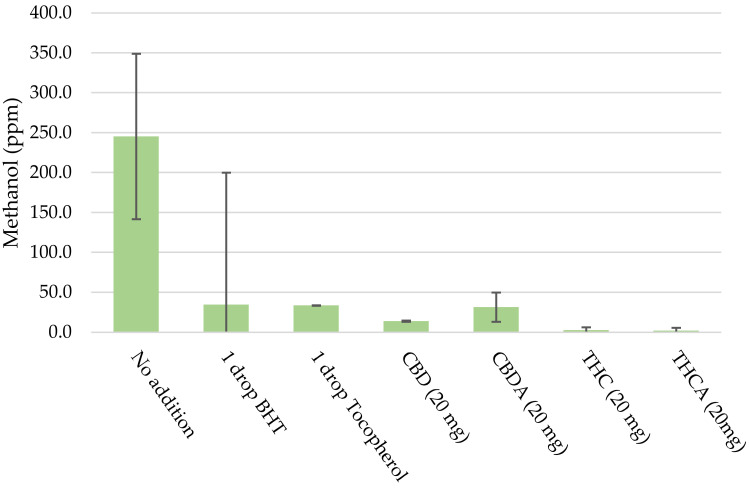
Detection of methanol (PPM) in samples with added antioxidants (error bars show the 95% confidence interval).

**Table 1 molecules-27-06037-t001:** Method parameters for the HS-GC/MS/MS method.

Gas Chromatograph		
Parameter	Data	
Column	Rxi-624Sil MS 30 m × 0.25 mm × 1.4 um	
Carrier gas	Hydrogen	
Flow rate	0.88 mL/min	
Mode	Constant pressure	
Split ratio	1:20	
**Temperature program GC**		
Rate (°C/min)	Temperature (°C)	Hold (minutes)
-	35	1.5
30	170	0
25	320	0.5
**Headspace Sampler**		
Oven temperature	140	°C
Sample line temperature	160	°C
Transfer line temperature	160	°C
Shaking	Off	
Vial Pressurization	25	PSI
Equilibration time	40.00	Minutes
Pressurization time	5.00	Minutes
Pressurization equilibration time	0.10	Minutes
Load time	0.20	Minutes
Load equilibration time	0.10	Minutes
Injection time	1.00	Minutes
Needle flush time	2.00	Minutes
**Mass spectrometer**		
Ion source temperature	260	°C
Transfer line temperature	310	°C
Acquisition mode	Selective Ion Monitoring	

**Table 2 molecules-27-06037-t002:** Analytes and their limit of detection (LOD), limit of quantification (LOQ), and action limits for levels in cannabis products.

Compound(s)	LOD	LOQ	BCC Action Limit
	PPM	PPM	PPM
1,2-Dichloro-ethane	0.1	0.3	1.0
Acetone	5	16	5000
Acetonitrile	0.8	2.5	410
Benzene	0.04	0.10	1.0
Butane	5.5	17.0	5000
Chloroform	0.25	0.50	1.0
Ethanol	0.5	1.0	5000
Ethyl acetate	31	84	5000
Ethyl ether	6	18	5000
Ethylene oxide	0.5	1.0	1.0
Heptane	33	90	5000
Isopropanol	7	21	5000
Methanol	62	167	3000
Methylene chloride	0.5	1.0	1.0
n-Hexane	0.5	1.0	290
Pentane	6.4	19.0	5000
Propane	20	59	5000
Toluene	12.2	33.0	890
Trichloroethene	0.5	1.0	1.0
Xylenes	3	10	2170

## Data Availability

Data will be attached during submission of the publication. We have not established a location for hosting. Guidance by the journal would be appreciated.

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
