# Peer review of "Acetone as Artifact of Analysis in Terpene Samples by HS-GC/MS"

_molecules, 2022, doi:10.3390/molecules27186037_

Round 1

Reviewer 1 Report

Reviewer Comments:

This paper is regarding the investigation of the cause of elevated acetone levels in neat terpene mixtures. To conclude, this paper needs to revise it carefully before it can be considered in impact journal like Molecules. Hope below comments will able to help to further improve the paper

·       please make sure that the paper is checked by native English speaker, the language needs improvement.

·       Please check Guides for Authors to make sure it is followed

Abstract:

·       Needs minor revision prior to the amendment of the main content.

·       An abstract is often presented separately from the article, so it must be able to stand alone. Hence the problem statement, aim, novelty and results of the study has all included in.

Introduction:

·       This section can be improved following the structure below:

1st paragraph: Problem statement

2nd paragraph: Current ongoing solution

3rd paragraph: Proposed solution in this work.

4th paragraph: Summarized the current research novelty and objective of this work.

·       Kindly refer below papers as it is highly relevant to this report:

o   Cannabis Sativa L.: A comprehensive review on the analytical methodologies for cannabinoids and terpenes characterization

o   Recent ultrasound advancements for the manipulation of nanobiomaterials and nanoformulations for drug delivery

Results and discussion:

·       Line 154-156; “This is likely due…” Please provide citation to justify this statement/explanation.

·       Please subscript “2” in “O2

·       Discussion needs improvement. Most of the explanation are not convincing, using the term “likely”. Please read more paper and justify your explanation with citation from journal articles.

·       Author also suggest to read this paper to enhance the scientific contain of this work, eg. The analytical landscape of cannabis compliance testing

Materials and methods:

·       The font of Table 1 & 2 is not consistent with the manuscript.

·       Equipment used in this research; please provide details in this form “Equipment (Model, manufacturer, Country)”

Conclusion

·       This section is missing

References

·       References format is consistent with the author guideline of Molecules journal. Good.

Papers suggested for reading:

o   Surface tuning of silica by deep eutectic solvent to synthesize biomass derived based membranes for gas separation to enhance the circular bioeconomy

o   Microalgae cultivation in wastewater and potential processing strategies using solvent and membrane separation technologies

Author Response

All authors are native English speakers and we are certain there is no major issue with the language. The language comment might have been marked by accident. 

The abstract was revised as suggested and the introduction was restructured as well. 

Suggested papers by the reviewer that were relevant to cannabis are now included as references. 

Experiment 3 was removed which improves the discussion. 

The reviewer states that the conclusion section is missing, but journal guidelines state the following:

"This section is not mandatory but can be added to the manuscript if the discussion is unusually long or complex."

Other comments have been addressed as well. 

Reviewer 2 Report

Manuscript Number: molecules-1894518

Journal: Molecules

Dear editor;

The manuscript entitled “Residual solvents as artifacts of analysis in terpene samples by 2 HS-GC/MS” has been reviewed. In the current manuscript, the authors aimed to investigate the cause of the high acetone levels reported in pure terpene mixtures. The manuscript is well-recognized, and the topic is good, this manuscript can be considered as a valuable contribution, but requires a major revision. I would like to raise some comments, and recommendations about the manuscript:

1. The abstract should be improved, and the best results achieved should also be indicated in numerical form. Also, the objectives should be clarified.

2. The introduction generally is good, with only a few grammatical errors. I also suggest that the authors clarify the objectives of the study and show their importance.

3. In the Materials and Methods section, many of the methods are not clearly described, and also the methodology requires the addition of a subsection dedicated to statistical analysis, which is extremely important for the interpretation of the results obtained.

4. The "Results" section needs to be considerably improved, especially with regard to the statistical analyses that are necessary to interpret and discuss the results scientifically.

5. All tables and figures missing the results of statistical analysis. The meaning of the error bars must be indicated.

6. The discussion section requires a very good discussion using recent references in the relevant fields and indicating the significance of the results compared to those in the literature.

7. The conclusion needs to be improved and the importance of the results achieved should be added.

8. The references need to be improved by using recent references.

Best regards

Author Response

  1. Abstract has been rewritten and numerical data has been added

2. Introduction has been rewritten and shortened significantly with the purpose to make the goal of the experiments more clear. 

3. Statistical section has been added. It would be helpful if the reviewer can explain what part of the experimental methods are not clear to him. Other reviewers did not have this comment. 

4+5. The explanation of the error bars was added. They are the 95% confidence interval of the data based on the replicate analysis. Where bars overlap there is no statistical significant difference.  Where bars do not overlap, there is statistical significant difference at the 95% level (p=0.05). This should address these two points. 

6. Reference added to the discussion section. There are no other papers raising the issue of acetone formation from terpenes as a source of failing testing for residual solvents. 

7. The introduction has been rewritten which should make the relevance of the conclusion more clear. 

8. References have been updated to more recent papers. 

Reviewer 3 Report

The article is devoted to the determination of the content of residual solvents in terpenes and their mixtures by HS-GC/MS. The data obtained could be useful for quality control of cannabis products. However, various decomposition processes for unsaturated hydrocarbons at high temperatures in the presence of oxygen are not new. And the only conclusion of this study, in my opinion, is that it is incorrect to determine the content of residual solvents in terpenes by HS-GC/MS in the presence of oxygen, which, given the nature of these compounds, is quite obvious. For a full-fledged study, it would be necessary to determine the final products of the degradation of these terpenes in addition to acetone (or methanol for linalool) and propose probable mechanisms for these degradations In addition, a number of other shortcomings can be noted:

1. Abstract and introduction are too long and contain a lot of information that has little to do with the title of the article

2. It would be useful to add the structures of the compounds under consideration

3. The title of the article refers to solvents, but in the following, only acetone and methanol (in the case of linalool) are discussed

4. The increase in the content of α-pinene in experiment 3 looks strange and is not explained

Given the above, I recommend rejecting the article

Author Response

The purpose of this study is not to determine the complete reaction schedules and how acetone might form from these compounds. The goal is to point out an issue with current methods used by commercial laboratories in the cannabis industry with certain non-cannabinoid containing samples. The fact that commercial laboratories are actually rejecting batches of terpene blends because of acetone levels above the action limit shows the relevance of the paper as well as the lack of obviousness to the operators of these facilities. To our knowledge there is no other publication linking the generation of acetone from solvents to the failing of residual solvent tests. Nor has anyone previously shown that antioxidants or cannabinoids are able to prevent this. 

  1. Abstract has been scrubbed and strongly reduced in length.
  2. Structures have been added
  3. Title has been changed to focus on acetone only
  4. Experiment 3 has been removed from the paper. 

Round 2

Reviewer 1 Report

I'm fine with the revision

Reviewer 3 Report

Although the manuscript has been revised and redundant off-topic information has been removed, it still lacks clarity about how this study might be useful for the analysis of herbal mixtures. In their response, the authors state the results of their work more clearly than in the abstract and introduction. In my opinion, the manuscript still does not correspond to the level of the journal. I would recommend revising the presentation of the findings and submitting the manuscript to a more suitable journal.